# Heart Disease and Arboviruses: A Systematic Review and Meta-Analysis

**DOI:** 10.3390/v14091988

**Published:** 2022-09-08

**Authors:** Jandir Mendonça Nicacio, Orlando Vieira Gomes, Rodrigo Feliciano do Carmo, Sávio Luiz Pereira Nunes, José Roberto Coelho Ferreira Rocha, Carlos Dornels Freire de Souza, Rafael Freitas de Oliveira Franca, Ricardo Khouri, Manoel Barral-Netto, Anderson da Costa Armstrong

**Affiliations:** 1Faculty of Medicine, Federal University of Vale do São Francisco—UNIVASF, Petrolina 56304-917, PE, Brazil; 2Postgraduate Program in Human Ecology and Socio-Environmental Management, Bahia State University—UNEB, Juazeiro 48904-711, BA, Brazil; 3College of Pharmaceutical Sciences, Federal University of Vale do São Francisco—UNIVASF, Petrolina 56304-917, PE, Brazil; 4Postgraduate Program in Applied Cellular and Molecular Biology, University of Pernambuco—UPE, Recife 50100-010, PE, Brazil; 5Oswaldo Cruz Foundation/Fiocruz, Institute Aggeu Magalhães, Recife 50740-465, PE, Brazil; 6Oswaldo Cruz Foundation/Fiocruz, Institute Gonçalo Moniz, Salvador 40296-710, BA, Brazil; 7Department of Medicine, Federal University of Bahia—UFBA, Salvador 40110-909, BA, Brazil; 8Rega Institute for Medical Research, KU Leuven, 3000 Leuven, Belgium; 9Instituto Nacional de Ciência e Tecnologia de Investigação em Imunologia, University of São Paulo, São Paulo 05347-902, SP, Brazil

**Keywords:** arboviruses, cardiac involvement, dengue, chikungunya, zika

## Abstract

Dengue fever, chikungunya, and zika are highly prevalent arboviruses transmitted by hematophagous arthropods, with a widely neglected impact in developing countries. These diseases cause acute illness in diverse populations, as well as potential cardiovascular complications. A systematic review was carried out to investigate the burden of cardiac involvement related to these arboviruses. Multiple databases were searched for articles that investigated the association of cardiovascular diseases with arboviruses, published up to March 2022. Relevant articles were selected and rated by two independent reviewers. Proportion meta-analysis was applied to assess the frequency-weighted mean of the cardiovascular findings. A total of 42 articles were selected (*n* = 76,678 individuals), with 17 manuscripts on dengue and 6 manuscripts on chikungunya undergoing meta-analysis. The global pooled incidence of cardiac events in dengue fever using a meta-analysis was 27.21% (95% CI 20.21–34.83; *I*^2^ = 94%). The higher incidence of dengue-related myocarditis was found in the population younger than 20 years old (33.85%; 95% CI 0.00–89.20; *I*^2^ = 99%). Considering the studies on chikungunya (*n* = 372), the global pooled incidence of cardiac involvement using a meta-analysis was 32.81% (95% CI 09.58–61.49, *I*^2^ = 96%). Two Zika studies were included that examined cases of infection by vertical transmission in Brazil, finding everything from structural changes to changes in heart rate variability that increase the risk of sudden death. In conclusion, cardiac involvement in arboviruses is not uncommon, especially in dengue fever.

## 1. Introduction

Dengue fever, chikungunya, and Zika are a group of acute febrile viral diseases transmitted by hematophagous arthropods that have afflicted diverse populations for decades, with the first descriptions of isolated urban outbreaks before 1960 [1,2]. Throughout history, these viruses have circulated in enzootic cycles, coexisting among wild animals and arthropod vectors in forests. However, epizootic cycles have become more frequent, affecting domestic animals, peridomestic vectors, and humans [3,4]. These viruses are transmitted to human hosts through the bite of infected female *Aedes* (*A.*) spp. mosquitoes, specifically *Aedes albopictus* and *Aedes aegypti*, which have different geographic distribution [5,6]. Population growth, climate change, accelerated destruction of biomes, and urbanization has contributed to virus spreading, setting these diseases permanently in the urban environment with devastating consequences on the affected population [2,3,7].

The dengue virus (DENV), responsible for dengue fever, is the most globalized arbovirus, with dramatic growth over the last 50 years. Before 1970, it was restricted to as few as five countries. Currently, about half of the world’s population is at risk of dengue infection, with estimates of 400 million people infected and 22,000 deaths annually [8,9]. Although the majority of infected individuals are asymptomatic or experience only a benign febrile illness, a minority of the individuals develop a life-threatening syndrome, known as severe dengue or dengue hemorrhagic fever. Among the symptomatic dengue cases, most of the infections lead to a self-limited acute febrile disease that lasts 1 to 2 weeks; however, a small proportion of the cases may evolve to the more severe disease. During the acute phase of the disease, nonspecific symptoms are commonly reported, including myalgia, arthralgia, headache, rash, nausea, and vomiting. Minor hemorrhagic manifestations such as petechiae and epistaxis may also occur. As described here, dengue disease manifestations are diverse. Thus, in 2014, the World Health Organization updated the clinical classification of dengue fever as follows: dengue without alarm signs, dengue with alarm signs, and severe dengue [10,11]. Severe dengue is defined by the presence of one or more of the following signs: plasma leakage that may lead to shock (dengue shock) and/or fluid accumulation with or without respiratory distress, and/or severe bleeding, and/or severe organ impairment. In severe dengue, specific organ impairment can occur without shock or any other features of severe dengue. A minority of these cases can progress to severe atypical manifestations such as encephalopathy, encephalitis, fulminant hepatitis, and myocarditis. Progression to severe dengue commonly occurs after the febrile phase, between days 4 and 6 of illness and may be fatal [3,12]. 

The chikungunya virus (CHIKV) is a mosquito-transmitted alphavirus that causes acute fever and acute and chronic musculoskeletal pain in humans. The virus was first described during the early 1950s in the Makonde Plateau, Newalla District, in the southern province of Tanzania [13]. Nowadays, CHIKV outbreaks are reported mainly in Asia and the Americas; Brazil being the most affected country in the Americas [14]. A chikungunya infection is characterized by a high attack rate that can reach 70%. The disease is most often characterized by an acute onset of fever (typically > 39 °C); other symptoms such as arthralgia, headache, myalgia, maculopapular rash, back pain, low back pain, and edema of the extremities are frequently reported. Unlike other arboviral diseases, this arbovirus infection is very symptomatic in most people, especially in the acute phase, but may also be symptomatic in the subacute and chronic phases, especially with joint manifestations. [15]. However, polyarthralgia is recurrent in 30–40% of infected individuals and may persist for years, especially in the elderly and patients with comorbidities [6,16,17,18]. Following infection, CHIKV replicates in the skin and then disseminates to the liver and joints. Moreover, samples from CHIKV-infected patients with myositic syndrome showed the presence of CHIKV antigens in skeletal muscle satellite cells [19]. Atypical manifestations are extra-articular and can affect any age group. Among these manifestations, neurologic complications such as encephalitis, meningitis, and Guillain–Barré syndrome, and cardiac complications such as arrhythmias, pericarditis, and myocarditis stand out. However, other manifestations may also occur, such as changes in the gastrointestinal tract, liver, kidneys, vesiculobullous skin lesions, and hematologic cells [15,16,20]. Detection of viral RNA in cerebrospinal fluid (CSF) samples of patients presenting neurological diseases reinforced the findings that the CHIKV infection is associated with Central nervous system (CNS) commitment [21].

The Zika virus (ZIKV) is a flavivirus, with contemporary outbreaks first recorded in 2007 in Oceania [22]. With the appearance of microcephaly in neonates in 2016 in Brazil, it has become, according to the World Health Organization, a “public health emergency of international concern”. By 2019, nearly 90 countries in various world regions had recorded autochthonous transmission of the Zika virus [23]. ZIKV commonly reported symptoms include rash, low grade fever (37.4 °C–38.0 °C), arthralgia, myalgia, fatigue, headache, and conjunctivitis. The most prominent clinical manifestation of ZIKV is microcephaly, a condition defined by an occipital-frontal head circumference (OFD) two standard deviations (SD) smaller than the average expected for age, gender, and population that was recently associated with a prenatal ZIKV infection [24]. Severe neurologic sequelae have also been described in adults, including meningitis, meningoencephalitis, and Guillain–Barre syndrome [25]. In addition, atypical and severe clinical manifestations may occur with myocarditis, pericarditis, heart failure, and arrhythmias such as atrial fibrillation, which are often underdiagnosed and increase mortality, especially in outbreak areas [26].

Despite international collaborative initiatives to contain neglected tropical diseases, there has been an apparent increase in the number of cases worldwide [27]. In Brazil alone, from January to May 2022, dengue, chikungunya, and Zika incidences increased more than 150%, 74%, and 214%, respectively, when compared to the same period in 2021 [28]. The increase in arbovirus infections generates a greater risk of complications or severe disease, especially in older populations and those with comorbidities [29,30]. Although reports exist showing severe cardiac abnormalities related to arbovirus infections, their actual frequency is still poorly known and varies by study [30,31,32]. This study aimed to carry out a systematic review of the literature regarding arbovirus-related cardiac complications, describing cardiac involvement in dengue, chikungunya, and Zika virus infections. In addition, a meta-analysis was performed to describe the incidence and estimate the risk of cardiac involvement in dengue fever.

## 2. Materials and Methods

### 2.1. Search Strategy

The systematic search was conducted in MEDLINE/PubMed, LILACS, Embase, Scopus and Web of Science, using MeSH and Entrees terms for PubMed and Embase, and DeCS (Health Sciences Descriptors) for the other databases. Two independent reviewers (JMN and JRCFR) with expertise in the topic performed the search on 1 March 2022, with the following strategy: (heart OR “heart disease” OR “heart attack” OR “heart failure” OR “cardiovascular disease” OR heart OR cardiovascular OR “cardiovascular disease”) AND (arboviruses OR dengue OR denv OR chikungunya OR chikv OR zika OR zikv). The Preferred Reporting Items for Systematic Reviews and Meta-Analyses (PRISMA) statement was used to conduct and report this systematic review (PRISMA, 2009). The studies were also selected using StArt (State of the Art through Systematic Review) computer software, version 2.3.4.2, from the Software Engineering Research Laboratory of the Federal University of São Carlos, São Paulo, Brazil (LAPES). This systematic review was registered in the International Prospective Register of Systematic Reviews (PROSPERO) as CRD42020219102.

### 2.2. Study Selection

Distinct study designs (case-control, cohort, prospective, and retrospective studies) involving cardiac complications in dengue, chikungunya, or Zika were selected. Non-original studies, reviews, editorials, abstracts, case reports, and animal model studies were excluded. There was no language limitation.

For study selection, two phases were considered: the first consisted of reading the titles and abstracts. After this phase, the selected full manuscripts were obtained for reading. Two independent reviewers performed both phases. Discrepancies in both phases were resolved through discussion between the reviewers or by a third reviewer (SLPN). The original study investigators were contacted to clarify data when necessary. Any disagreements about study inclusion were resolved by consensus or arbitration by a third reviewer (SLPN).

The methodological quality instrument NOS was applied and cohort studies were classified according to their methodology. Three criteria are included in the NOS checklist: study selection (maximum 4 stars), comparability (maximum 2 stars), and assessment of outcome (maximum 3 stars), with researchers rating each criterion independently [33]. Manuscripts above six stars were considered good quality and low risk of bias; those with six stars were considered moderate quality; and those with five stars or less were considered low quality and, therefore, high risk of bias studies (Appendix A).

### 2.3. Data Collection and Extraction

After the final selection of the research articles, a form was created during a meeting between the researchers to extract the data from the selected studies for subsequent analysis of the results and discussion. Data were extracted based on patient characteristics, age group, sex, clinical classification of arboviruses, cardiac involvement, cardiac tests performed, comorbidities, study quality, and inclusion and exclusion criteria. The agreement was quantified by the kappa statistical method [34]. Levels of evidence were attributed according to the Oxford Centre for evidence-based medicine [35,36]. The quality of each study was assessed with the Newcastle–Ottawa Scale (NOS) [33]. The Meta-Analyses of Observational Studies in Epidemiology (MOOSE) checklist was applied to assess and reduce bias in the data analysis [35].

### 2.4. Statistical Analysis

Statistical analysis was performed using the software RStudio, version 4.2.0, 2022.02.2+485 (Boston, MA, USA), Open Source License, using the meta package (General Package for Meta-Analysis, 2022). Due to methodological limitations and the low number of included manuscripts, meta-analysis was not performed for the studies on chikungunya and Zika.

We conducted a meta-analysis of proportions to assess the global incidence of cardiac involvement in patients with dengue fever. The results were presented as pooled proportions (%) with a 95% confidence interval (CI). Possible publication bias regarding the incidence of cardiac involvement in patients with arboviruses was evaluated using Begg’s and Egger’s tests combined with a funnel plot [37].

To prevent misleading conclusions, the Freeman–Tukey double arcsine method was used to stabilize the variance [38]. Heterogeneity and consistency were evaluated using Cochran’s Q test and *I*^2^ statistics. The magnitude of heterogeneity was tested by the *I*^2^ method, presented in percentage values of the variance, ranging from 0% to 100%, and observing the actual size effect from all studies considering values above 50% as substantial heterogeneity and values above 70% as high heterogeneity [39,40]. Subgroup meta-analysis was used to assess possible sources of heterogeneity: age group, continent, and study country. Meta-regression, when applicable, was performed using random-effects analysis on covariates. In these analyses, *p*-values less than 0.05 were considered statistically significant. 

## 3. Results

Of the 4209 publications identified using the search strategies described above, 42 articles were selected after meeting the inclusion criteria. The verification of cited articles and citation of included articles yielded nine relevant observational, prospective, and retrospective descriptive studies [41,42,43,44,45,46,47,48,49]. The study selection flowchart for the different phases of the systematic review is displayed in Figure 1.

After applying the inclusion and the exclusion criteria, we included 42 studies, distributed as follows: 34 articles on dengue [31,41,42,43,44,45,46,47,48,49,50,51,52,53,54,55,56,57,58,59,60,61,62,63,64,65,66,67,68,69,70,71,72,73]; 6 studies on chikungunya [32,74,75,76,77,78]; and 2 on Zika [79,80]. Two studies used a control group of people exposed to arboviruses [32,47]. The kappa test showed high reliability between reviewers (k = 0.886, *p* = 0.00), with an agreement percentual of 94.3% [33,81]. 

### 3.1. Study Quality

A total of 52% of the studies (22 studies) presented acceptable methodological quality. Regarding the level of evidence, as these are cohort studies, the level of evidence was considered IIb [34]. Observational prospective studies on dengue that presented appropriate data (similar methodological designs) were included in the meta-analysis. It was impossible to perform a meta-analysis for the studies on Zika. No apparent asymmetry in the funnel plot was observed and the absence of evidence of suspected publication bias was supported by Begg’s statistical test (*p* = 0.423). Asymmetry in the funnel plot was observed by Egger’s statistical test (*p* = 0.001). A funnel plot was generated for the results with data from 34 studies on dengue, applying adjustment by the trim-and-fill method to assess the possible publication bias (Appendix A).

### 3.2. Data Synthesis

A total of 42 articles were selected, resulting in 76,678 individuals exposed to arboviruses; 76,188 were infected by dengue, 372 by chikungunya, and 118 by Zika. The majority of the study participants (51%) were males. Only two manuscripts were published in Spanish, all others in English.

Among the 34 manuscripts on dengue, 32 were prospective studies and 2 were retrospective. As for the six articles on chikungunya, three was prospective and the other three were retrospective. There were only two Zika studies, one retrospective and one prospective. The majority of the studies came from the continent of Asia, with 13 from India, 6 from Thailand, 3 from Sri Lanka, 3 from Taiwan, and 1 from each of the following countries: China, Vietnam, Pakistan, Indonesia. There were also two manuscripts from Colombia, four from Brazil, and one article from each of the following countries: France, Paraguay, Cuba, French Guiana, Puerto Rico, and Guadalupe (Table 1).

Of the 42 selected studies, 38 were single-center, with diagnoses by clinical-epidemiological criteria and laboratory confirmation. Of these, five included in their casuistic only severe dengue cases [41,58,61,77,78]. Thirty-eight studies used convenience sampling, two used sample size calculation [44,49], and one used the purposive sampling method [68]. Among the 42 selected studies, 11 included patients under 20 years old and children (1024 participants). Although 65% of the selected studies were published after 2015, most used clinical classification criteria for dengue severity before the 2014 update by the World Health Organization (Table 1). 

Only 24% of the studies reported the prevalence of comorbidities and 20% of the studies did not include participants with comorbidities. The remaining 63% studies did not provide information on comorbidities. Regarding the clinical classification of arboviruses, 64% of the studies reported greater cardiac involvement in the more severe forms of the disease and 20% did not provide this information. It is important to note that the selected articles comprised a period from 1973 to 2022, involving different classifications for the severity of arboviruses. Only 8.3% of the manuscripts reported secondary dengue infection, without making clear its potential association with cardiac involvement (Table 1).

Among the papers on dengue, 72% identified electrocardiographic abnormalities in their samples, with sinus bradycardia and tachycardia being the most frequent findings. Left ventricular dysfunction identified by echocardiogram (ejection fraction < 50%) and changes in cardiac biomarkers (CK-MB, troponin, NT-pro BNP) were recorded in 199 and 1066 subjects, respectively.

### 3.3. Studies on Zika

Although there are several articles in the literature that address cardiac involvement related to ZKV (case report, editorial, abstract, cross-sectional study, reviews), we identified two articles in this review that met the inclusion and exclusion criteria up to the time of the search. These were observational studies (retrospective and prospective), evaluating cases of infection by vertical transmission in Brazil with a mean age between 58 days to 16 months. In one of the studies, a retrospective analysis of newborns in northeastern Brazil found that more than 13% of the sample had congenital cardiac abnormalities associated with the Zika virus (ostium secundum, a small apical muscular ventricular septal defect), although they have been little studied. Another prospective study showed a statistically significant difference between R-R values in patients with congenital Zika virus syndrome (24-h Holter monitoring), which may be associated with the risk of sudden infant death syndrome, suggesting early surveillance of these children [79,80]. The lack of more consistent primary studies on this scenario so far compromises the findings on cardiac outcomes in adults and children.

### 3.4. Meta-Analyses of Studies on Chikungunya

Because of the small number of studies identified for chikungunya in this review, we performed meta-analytic estimates and calculated the effect measure (weighted mean incidence) of all studies found, i.e., six studies, three prospective, and three retrospective, with 52% of the sample consisting of men. The global pooled incidence of cardiac events using the meta-analysis of the random-effects model was 32.81% (95% CI 09.58–61.49, *I*^2^ = 96%, *p* < 0.01, 06 studies, 372 patients) (Figure 2).

Subgroup analysis by primary study design showed a pooled incidence of 33.75% for prospective studies and 31.92% for retrospective studies, maintaining high heterogeneity (*I*^2^ = 96%, *I*^2^ = 98%, respectively). The main cardiac manifestation described was cardiovascular failure, which was associated with atypical and severe forms of CHIKV, especially severe sepsis or septic shock, in more than 80% of cases. Regarding myocarditis, the average incidence was low (2.38%; 95% CI 0.00–09.37, *I*^2^ = 88%, *p* < 0.01, six studies, 372 patients); however, information on the type of cardiac events was lacking in the primary studies analyzed. In addition, comorbidities were reported in more than 90% of cases, with a predominance of hypertension, diabetes mellitus, kidney disease, ischemic heart disease, and chronic heart disease (Table 1).

### 3.5. Meta-Analyses of Studies on Dengue

Only observational prospective single-center studies on dengue with NOS ≥ 6 were used to perform meta-analytic estimates and to calculate the effect measure (weighted mean incidence), i.e., 17 studies: 13 in patients 20 years of age or older and 4 in patients under 20 years. The global pooled incidence of cardiac events using the meta-analysis of the random-effects model was 27.21% (95% CI 20.21–34.83, *I*^2^ = 94%, *p* < 0.01, 17 studies, 4616 patients). Subgroup analysis by age showed a pooled incidence of 28.32% for 20 years or older and 24.31% for under 20 years, maintaining high heterogeneity (*I*^2^ = 80%, *I*^2^ = 97%, respectively) (Figure 3).

Subgroup analysis by continent showed a pooled incidence of 31.27% in Asia (95% CI 24.30–38.70, *I*^2^ = 87%, *p* < 0.01, 14 studies, 3698 patients) and 12.34% in South America (95% CI 2.53–27.92, *I*^2^ = 97%, *p* < 0.01, 2 studies, 737 patients) (Figure 4).

Similarly, to improve the accuracy of the results, a subgroup meta-analysis was performed by countries with at least two studies. The analysis of studies carried out in India showed a pooled incidence of 31.17%, with lower heterogeneity and greater consistency of results (95% CI 26.34–36.22, *I*^2^ = 68%, *p* < 0.01, seven studies, 1187 patients) (Appendix A). 

The pooled incidence of cardiac events in patients with dengue over 20 years of age by altered ECG was 27.60% (95% CI 19.75–36.21, *I*^2^ = 97.1%, *p* < 0.01, 12 studies, 3856 patients), with sinus bradycardia and sinus tachycardia found in 11.37% (95% CI 7.90–13.34, *I*^2^ = 84.5%, *p* < 0.01, 10 studies, 1912 patients) and 7.41% (95% CI 3.43–12.65, *I*^2^ = 91.8%, *p* < 0.01, 9 studies, 1485 patients), respectively. In this same group of adults, the pooled incidence for ejection fraction < 50% was 3.42% (95% CI 1.03–6.95, *I*^2^ = 90.4%, *p* < 0.01, 8 studies, 1365 patients); for elevated troponin it was 12.44% (95% CI 5.41–21.74, *I*^2^ = 97.6%, *p* < 0.01, 11 studies, 3288 patients); for elevated CK-MB it was 3.42% (95% CI 1.03–6.95, *I*^2^ = 90.4%, *p* < 0.01, 8 studies, 1365 patients); for myocarditis it was 10.94% (95% CI 4.55–19.55, *I*^2^ = 96.8%, *p* < 0.01, 11 studies, 3605 patients). When evaluating myocarditis by age subgroup, a higher incidence was found in the population under 20 years, 33.85% (95% CI 0.00–89.20, *I*^2^ = 99.0%, *p* < 0.01, three studies, 298 patients). Nevertheless, when evaluating cardiac events in the studies that used the same diagnostic (European Society of Cardiology, 2013), a pooled incidence of 27.13% was found in adults, whose result was statistically consistent despite the small sample and different demographic characteristics (95% CI 25.28–29.03, *I*^2^ = 0.0%, *p* = 0.56, four studies, 2195 patients) (Figure 5).

### 3.6. Meta-Regression 

Due to the high heterogeneity already expected for this meta-analysis of observational studies, a random-effects meta-regression was performed, considering variables of interest (sex, year of study publication, clinical severity of dengue, death associated with cardiac event). The regression model showed a good fit (t^2^ = 0.022). The heterogeneous performance accounted for 92.09% of the residual variance, showing that the year of publication of the study was significantly associated with a reduction in cardiac events (*p* = 0.004; Figure 6). However, there was no association with cardiac outcome according to sex, lethal outcome, and clinical severity (Appendix A).

## 4. Discussion

Arboviruses are associated with acute infections that are increasingly frequent in urban populations, with cyclical outbreaks affecting more vulnerable populations. The presence of the mosquito vector, climatic conditions, and the fragility of public policies favor their dissemination [28]. We showed that cardiac involvement in arboviruses is not a rare complication. Particularly in dengue and chikungunya, the incidence of cardiac complications was found in over a quarter of infected participants. However, it is important to emphasize that this result refers to hospitalized patients with dengue fever or chikungunya and does not apply to the general population exposed to this arbovirus.

Little is known about the pathophysiology of cardiac lesions, however, with regard to dengue, it is related to disease severity and correlates with the extent of plasma leakage [66]. However, other mechanisms have been described, such as tropism of the virus for the myocardium, genetic factors, and a strong host inflammatory response leading to tissue destruction. Similar mechanisms of cardiac tropism have also been described in CHIKV and ZIKV, which trigger an intense inflammatory process with the release of proinflammatory cytokines (IL -18, TNF-α, IFN-γ) and damage cardiac tissue [26,66,79].

In this review, we found a high mean incidence of cardiac events (33%) in chikungunya patients, with more than 90% of these patients having at least one comorbidity (mainly hypertension and diabetes mellitus). Similarly, Alvarez et al., described cardiac involvement in 54.2% of cases in the populations of countries in the Americas, Asia, and Europe [82]. However, in contrast to other studies, we did not find a predominant myocarditis. One of the explanations for this finding would be the heterogeneity of the samples studied and, in particular, the lack of reporting of the nature of cardiac events (e.g., the causes of heart failure were not detailed), so it was not possible to extrapolate this result to the general population. 

To justify the involvement of the heart in the context of these arboviruses, there is strong evidence that the tropism of CHIKV applies not only to fibroblasts and interstitial connective tissue but also to several other organs and tissues such as the spleen, skin, lung, bone, liver, and skeletal muscle. In the heart, a CHIKV antigen has been detected in fibroblasts, vascular endothelium, myocardium, and adipose tissue [75].

To date, there is a lack of prospective studies with acceptable methodological quality, especially concerning chikungunya and Zika, which compromises the most consistent epidemiological evaluation. Recent systematic reviews of cardiac complications in patients with dengue, chikungunya, and Zika have included primary studies of poor methodological quality as case reports in 30 to 50% of the sample [26,30,83]. Our review included only prospective and retrospective studies in order to bring more consistency of results. 

Based on the methodological proposal of this review, and within the time period that the databases were searched, it was not possible to include studies with cardiac involvement in adults. Nevertheless, the two included articles address an understudied complication (congenital heart defects and changes in heart rate variability) that increases the risk of sudden death in these children. This is particularly important in countries where there is a high risk of an outbreak of this disease, such as Brazil, which increases exposure of pregnant women to ZKV and, consequently, vertical transplacental transmission, which can lead not only to microcephaly but also to cardiac involvement [26,28]. 

When assessing the pool of dengue-related cardiac abnormalities, myocarditis appears to be the most common potentially harmful finding. Although most of the myocarditis findings are subclinical with enzymatic and electrocardiographic changes, we found that about 5% of those with dengue fever had left ventricular dysfunction. Also associated with low left ventricular fraction, biomarkers (CK-MB, troponin) have been frequently described in observational studies and they are considered to have a poor prognosis when they appear together. In this scenario, recent studies have shown that elevated troponin is more sensitive in identifying minor myocardial injuries [54,66]. 

Importantly, this review showed that younger people appear to be at a higher risk for dengue-related myocarditis. In general, myocarditis in children has been increasing in recent decades, with peaks in children under 2 years of age and adolescents, accounting for 5% of sudden cardiac deaths in this age group [84,85,86]. This phenomenon, although not well understood, is related to inflammatory factors, with the release of cytokines (TNF-α, interleukins 6, 13, and 18, and cytotoxic factor), greater susceptibility to viral infections with possible tissue infiltration, host genetic factors, and viral load [30,67,81,83]. Despite this, ventricular dysfunction associated with myocarditis and electrocardiographic changes are, in most cases, transient [55,62]. Therefore, it is clear that myocarditis is one of the most important complications in patients with dengue, especially in children and young adults. However, we lack controlled studies to understand the association of dengue and myocarditis better, allowing early diagnosis and treatment of this condition. 

When using meta-regression to evaluate possible sources of heterogeneity, we did not observe a significant association of cardiac events with dengue classification. However, case series and observational studies often show this association, probably due to the involvement of an overactive immune response, which can lead to myocarditis with left ventricular dysfunction [62,69]. This lack of association in the meta-regression is probably related to conflicting clinical classifications of these arboviruses over time. 

As a less potentially harmful finding, we report bradycardia as the most common arbovirus-related electrocardiographic abnormality. Previous data indicated a predominance of sinus tachycardia, especially in the acute phase [30]. This divergence may be related to the clinical phase of arboviruses, as some of the studies included in this review reported this finding during disease defervescence. The lack of information from the primary studies precluded analyzing this outcome in this meta-analysis.

Despite being well-known diseases, arboviruses challenge the health of populations mainly in metropolitan and peri-urban areas of tropical countries, due to reinfestation of their main vector (*Aedes aegypti*), causing increasingly severe clinical repercussions, as is the case of Zika in the Americas. In addition, these arboviruses (mainly dengue and chikungunya) have been reaching other ecological areas, such as temperate climate countries, through the uncontrolled expansion of another important vector (*Aedes albopictus*), generating large outbreaks and a sustained cycle of diseases, with serious health repercussions [87]. Therefore, a proper vector control policy is fundamental to minimize the impacts of this disease on populations.

Another important strategy to control symptomatic disease and to reduce its complications is to promote the training of healthcare teams for early detection and for monitoring of signs and symptoms. No less important, especially in endemic areas at a higher risk of outbreaks, is the development of vaccines. Although some vaccines are already commercially launched vaccines, their effectiveness is still considered low. Global collaboration is therefore needed to develop more effective vaccines, as was the case, for example, with COVID-19 [88].

This review has limitations due to the scarcity of primary studies with better methodological qualities, compromising the consistency of the effect size. There were information gaps, especially regarding comorbidities and secondary infections, in addition to the heterogeneity of the populations studied. Despite these limitations, this review brings an analysis of prospective primary studies with a better methodological quality, confirming that, through the estimation of grouped frequency, the cardiac involvement in arboviruses are not uncommon manifestations, being necessary preventive actions.

## 5. Conclusions

In conclusion, cardiac involvement is not uncommon as a complication of arboviruses, with higher quantity and quality data published for dengue fever. Myocarditis was the most frequent and potentially harmful cardiac complication in dengue fever, with indicatives of a higher burden in youth. 

## Figures and Tables

**Figure 1 viruses-14-01988-f001:**
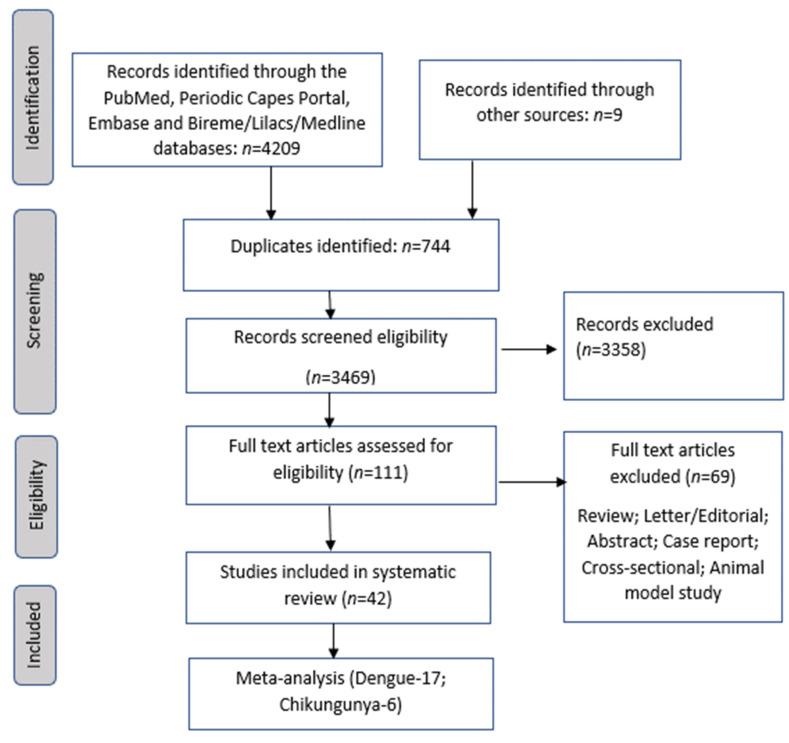
Flowchart of study selection for inclusion in the literature review.

**Figure 2 viruses-14-01988-f002:**
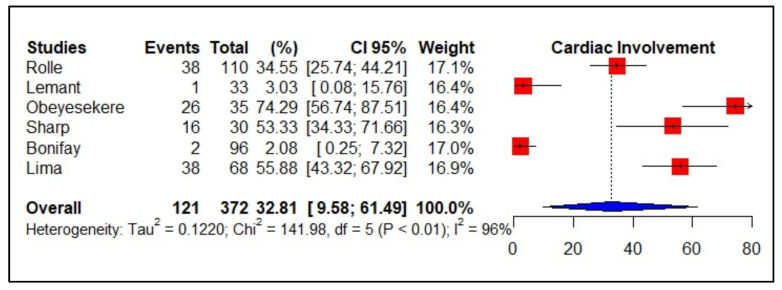
Forest plot comparison of the frequency of cardiac events in patients with chikungunya. **Legend**: CI—confidence interval.

**Figure 3 viruses-14-01988-f003:**
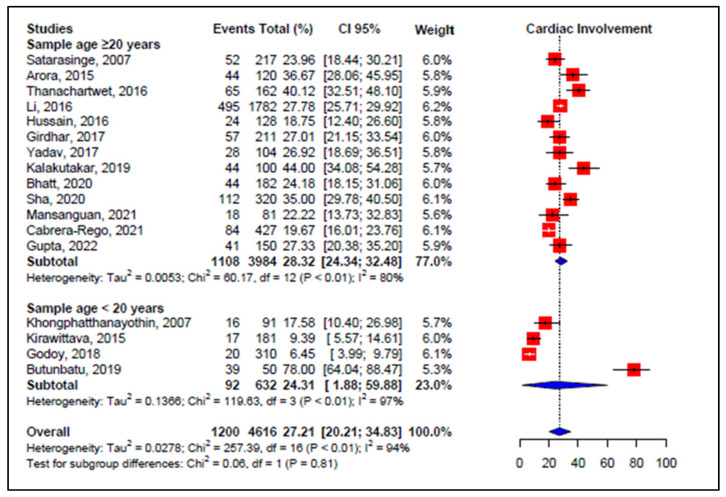
Forest plot comparison of the frequency of cardiac events in patients with dengue, grouped by age group. **Legend**: CI—confidence interval.

**Figure 4 viruses-14-01988-f004:**
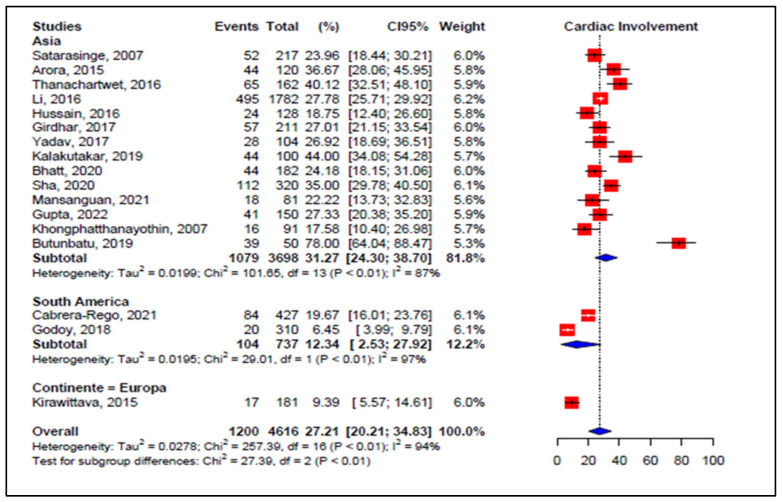
Forest plot comparison of the frequency of cardiac events in patients with dengue, grouped by continent. **Legend**: CI—confidence interval.

**Figure 5 viruses-14-01988-f005:**
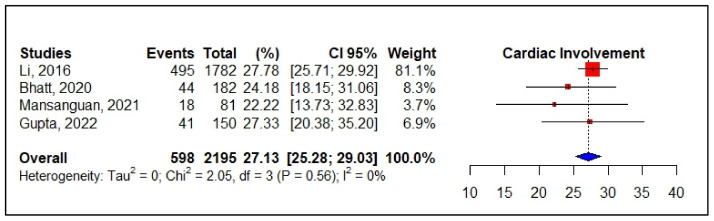
Forest plot of the frequency of cardiac events in dengue, using ESC criteria, 2013. **Legend**: CI—confidence interval; ESC—European Society of Cardiology.

**Figure 6 viruses-14-01988-f006:**
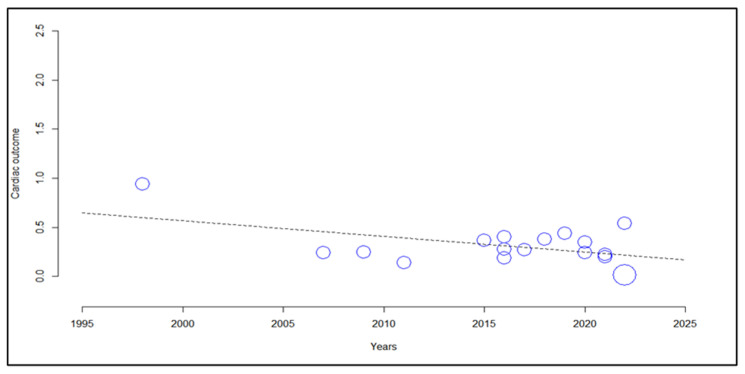
Scatter plot showing the frequency of cardiac events in patients with dengue through meta-regression.

**Table 1 viruses-14-01988-t001:** Summary of the included studies.

First Author Year Journal	Country /Region	Type of Study Follow Up	Study Group(*n*) Age/Sex(M/F)	Cardiovascular Findings	Severity Arboviruses (WHO)	Cardiac Involvement (*n*) Comorbidities
DENGUE						
Wali et al., 1998 [70] International Journal of Cardiology	India	Observational study, prospective Follow-up: 3 weeks	*n*: 17 M: 12/F: 5 Mean age: 29.7 yrs	EF 33–44%: 7 EF 41–50%: 9 Sinus bradycardia: 3	DSS-8 DHF-9 WHO, 1997	*n*: 16 Related to gravity Comorbidities: not reported
Kabra et al., 1998 [46] The National Medical Journal Of India	India	Prospective observational study Follow-up: 2 months	*n*: 54 Mean age: 6.3 yrs M: 22/F: 32	EF ^#^ < 50%: 9 (EF 35–50%: 7 EF < 35%: 2) First degree atrioventricular block: 1	Dengue fever, dengue, DHF *, and DSS ** WHO, 1986	*n*: 9 Not related severity Comorbidities: not reported
Khongphatthanaythin et al., 2003 [62] Intensive Care Med	Thailand	Prospective observational study Follow-up 2 weeks	*n*: 24 Mean age: 11 yrs M: 11/F: 13	EF lower during toxic stage	DHF: 24 Comorbidities: not reported WHO, 1997	*n*: 24 Amount of cardiac involvement not reported
Khongphatthanayothin et al., 2007 [57] Pediatr Crit Care Med	Thailand	Prospective observational study Follow up: 17.5 days	*n*: 91 Age: 5–15 years Male: 52/F: 39	EF < 50%: 16	Dengue fever-30 DHF-36 DSS-25 WHO, 1997	*n*: 16 Related to gravity Comorbidities: not
Satarasinghe et al., 2007 [58] British Journal of Cardiology	Sri Lanka	Observational, prospective study Follow-up: 3 months	*n*: 217 Age distribution: 12–65 yrs M: 144/F: 73	Myocarditis: 52 Sinus bradycardia: 52	Dengue classification not reported	*n*: 52 Not correlation severity Comorbidities: not reported
Wichmann et al., 2009 [47] South Asian J Trop Med Public Health	Sri Lanka	Hospital based observational descriptive study	*n*: 133 Age: 18–76 years M 69/F: 64	Myoglobin: 60 CK-MB: 17 Troponin-1 NT-Pro BNP: 25	Dengue fever Secondary dengue: 66 WHO, 1997	*n*: 33 Comordities: not reported
Salgado et al., 2010 [68] Pediatr Infect Dis	Colombia	Observational, prospective Follow-up: 1 year	*n*: 102 Mean age: 6 yrs M/F: not reported	Myocarditis: 11 Pleural effusion: 5 Sinus bradycardia: 9 Sinus tachycardia: 2 CK-MB: 6	DHF-79 Dengue fever: 23 WHO, 2009	*n*: 11 Related to gravity Comorbidities: not reported
La-Orkhun et al., 2011 [72] Annals of Tropical Paediatrics	Thailand	Prospective Follow up: by 14 days	*n*: 35 Mean age: 11.7 yrs M: 20/F: 15	First degree atrioventricular block: 2 First degree atrioventricular block: 3 Other abnormalities: 10	Dengue fever-12 DHF-18 DSS-5 WHO,1997	*n*: 15 Not related to gravity Comorbidities: not
Weerakon et al., 2011 [67] BMC Research	Sri Lanka	Observational prospective Follow up: not reported	*n*: 319 Age: 13–31 yrs	Death: 11 Myocarditis: 21 Pleural effusion: 41 Abnormal ECG: 5 Troponin T: 5	Dengue fever: 153 Severe dengue: 166 WHO, 1997 Secondary infection	*n*: 45 Related to gravity Comorbidities: not reported
Khositseth et al., 2012 [71] Journal of Pediatric Intensive Care	Thailand	Prospective observational study Follow up: 3–44 days	*n*: 8 M: 5/F: 3 Mean age: 6.5 yrs	Impaired systolic function: 1	Death: 06 DSS WHO, 1997	*n*: 1 (impaired systolic function) Comorbidities: not reported
Kumar Yadav et al., 2013 [63] Pediatr Cardiol	India	Prospective observational study	*n*: 67 Mean age: 10.4 yrs Male: 65%	Myocarditis-32 Pericardial effusion: 1 EF< 35%: 3 Tei index ^&^ abnormal: 48	Death: 01 Dengue severe Secondary WHO, 2009	*n*: 48 Related to gravity Comorbidities: not
Saldarriaga et al., 2013 [65] RevistaColombian de Cardiologí	Colombia	Observational descriptive prospective study	*n*: 7 M: 4/F: 3 Mean age: 55.7 yrs	Systolic dysfunction: 2 Pericardial effusion: 1	Not clinical classification. WHO, 2011	*n*: 3 Comorbidities: coronary heart disease and heart failure
Miranda et al., 2013 [73] Clinical Infectious Diseases	Brazil	Observational prospective descriptive study Follow up: 3–6 days	*n*: 81 M: 39/F: 42 Mean age: 32 yrs	Myocarditis: 3 Impaired systolic function: 4 Troponin: 6 NT Pro BNP: 10	Dengue fever-54 Dengue hemorragic fever-27 WHO, 1997	*n*: 12 Comorbidities: not reported
Arora et al., 2015 [48] JAPI	India	Prospective observational Follow up not reported	*n*: 120 Mean age-32 yrs M: 85/F: 35	Myocarditis: 45 Sinus bradycardia: 10 Sinus tachycardia: 4	Dengue fever-20 DHF-85 * DSS-15 ** WHO, 2009	*n*: 45 Related to gravity *Comorbidities*: not reported
Kirawittaya et al., 2015 [59] PloS NT	France	Observational analytical longitudinal prospective	*n*: 181 M/F: not reported Follow up: 1 week	EF < 56%: 17 Pericardial effusions: 7 Without myocarditis	Dengue fever DHF-23 WHO, 1997; 2009	*n*: 17 Related to gravity Comorbidities: not reported
Li et al., 2016 [50] Medicin	China	Observational analytical longitudinal prospective	*n*: 1782 M: 203/: 224 Mean age: 60.7 yrs	Myocarditis: 201 ST-T abnormality: 59 Atrial fibrillation: 26 Troponin I: 13 CK-MB: 54 NT Pro BNP: 81	DF *** without warning signs 1707 DF with warning signs +severe cases-75 WHO, 2009	*n*: 495 Related to gravity Comorbidities: not reported
Pothapregada et al., 2016 [31] Indian J Pediat	India	Observational retrospective study Follow up: 8.1 days	*n*: 254 Mean age: 6.9 yrs M: 132/F: 122	Myocarditis: 5 Diastolic dysfunction: 2 Pericardial effusion: 3 Paroxysmal supraventricular tachycardia: 3 Sinus bradycardia: 2	Death-6 Dengue fever-159 Severe dengue-95 WHO, 2011	*n*: 5 Related to gravity Comorbidities: not reported
Thanachartwet et al., 2016 [49] PloS One	Thailand	Prospective observational study Follow-up total: 15 days	*n*: 162 Median age: 24.5 yrs M: 87/F: 75	Abnormal ECG ^##^: 65 Troponin T: 2 NT Pro BNP: 23	Death: 02 DSS-17 Not DSS-145 WHO, 2009	*n*: 65 Related to gravity Comorbidities: not reported
Hussain et al., 2016 [51] P J M H S	Pakistan	Observational, prospective study. Follow up: two weeks	*n*: 128 Above 12 yrs	Myocarditis: 24	Death: 20 Dengue fever DHF and DSS-122 WHO, 2011	*n*: 24 Related to gravity Comorbidities: not
Girdhar et al., 2017 [55] Jour of Med Science and Clinical Search	India	Hospital based observational descriptive study	*n*: 211 M: 110/F: 101 Mean age: 30.4 yrs	Pericardial effusion: 10 EF ^#^ < 50%: 10 Sinus bradycardia: 18 Sinus tachycardia: 14 Troponin T: 12 CK-MB: 79	Dengue fever: (no clinical severity classification) WHO, 2009	*n*: 57 Related to gravity Comorbidities: not reported
Yacoub et al., 2017 [69] PLoS NTD English	Vietnam	Prospective observational study Follow up: 14 days	*n*: 102 Median age: 11 yrs F: 102	Left ventricular dysfunction: 24 Right ventricular dysfunction: 6	DSS-80 WHO, 2009	*n*: 30 Related to gravity Comorbidities: not reported
Yadav et al., 2017 [52] J. Evolution Med. Dent. Sci	India	Observational prospective descriptive study	*n*: 104 Age: 21–30 yrs	Left ventricular dysfunction: 4 Right ventricle dilation: 2 Sinus bradycardia: 9 Sinus tachycardia: 7 Troponin T: 9	Not clinical Dengue classification WHO, 2012	*n*: 28 Related to gravity Comorbidities: not
Lakshman et al., 2018 [41] Tropical Doctor	India	Prospective observational Follow up: not reported	*n*: 50 Median age: 38 yrs M: 35/F: 15	Left ventricular dysfunction: 8 Sinus tachycardia: 10 CK-MB: 10 Troponin I: 3	DF *** not warning-10 DF with warning-33 Severe DF-7 WHO, 2014	*n*: 16 Not related to gravity *Comorbidities*: not reported
Godoy et al., 2018 [60] Pediatr. (Asunción)	Paraguay	Prospective observational study Follow up: 2 weeks	*n*: 310 Mean age: 13 yrs M: 16/F: 29	Pericardial effusion: 2 Sinus bradycardia: 19 first degree atrioventricular block: 1	Dengue fever-12 DF with warning signs-8 WHO, 1997	*n*: 20 Related to gravity Comorbidities: not reported
Buntubatu et al., 2019 [61] Jour of Tropical Pediatrics	Indonesia	Prospective observational Follow-up: 2 to 7 days (median 3 days)	*n*: 50 Median age: 8 yrs M: 15/F: 35	Myocarditis: 39 Sinus tachycardia: 11 Sinus tachycardia: 4 CK-MB: 35 Troponin I: 12	Death: 0 Dengue fever-15 DHF-12 DSS-23 WHO, 2011	*n*: 39 Related to gravity Comorbidities: not reported
Kalakutakar et al., 2019 [64] Inter Journal of Current Microb	India	Observational prospective descriptive study	*n*: 100 M: 59/F: 41 Mean age: 30 yrs	Myocarditis: 2 Sinus bradycardia: 15 Sinus tachycardia: 9	Dengue fever: (no clinical severity) WHO, 2009	*n*: 44 Not studied relationship with clinical severity.
Bhatt et al., 2020 [53] Infection	India	Prospective observational Follow up: 7 days	*n*: 182 Mean age: 30 yrs M: 126/F: 56	Myocarditis: 13 EF < 50%: 11 Sinus bradycardia: 10 Sinus tachycardia: 30 Troponini I: 25 NT Pro BNP: 22	Death: 5 Dengue fever: 37 DF warning signs: 85 -DF severe: 60 WHO, 2014	*n*: 44 Related to gravity *Comorbidities*: not reported
Sha et al., 2020 [54] International Journal of Cardiology	India	Observational analytical longitudinal prospective	*n*: 320 M: 198/F: 122 Age: 18–80 yrs	Myocarditis: 56 Pleural effusion: 3	Death-14 Dengue classification not reported WHO, 2009	*n*: 112 Not related to gravity Comorbidities: not reported
Cabrera-Rego et al., 2021 [56] Enf Infec.Microb Clin	Cuba	Observational longitudinal prospective	*n*: 427 Age: <25–>65 yrs M: 237/F: 190	Myocarditis: 1 Pericarditis: 7 Sinus bradycardia: 59 Atrial fibrillation: 2	Dengue classification not reported Comorbidities: not WHO, 2009	*n*: 84 Related to gravity *Comorbidities*: not
Mansanguan et al., 2021 [44] BMC Infect Dis	Thailand	Observational prospective descriptive study Follow up: 2 weeks	*n*: 81 Mean age: 33 yrs	Myocarditis-2 Left ventricular systolic dysfunction: 3 Troponin: 2	Dengue fever: 39 Dengue hemorragic fever: 42 WHO, 1997	*n*: 18 Comorbidities: diabetes mellitus, hypertension Related to gravity
Lee et al., 2021 [71] Jour of Microb Immunology and Infection	Taiwan	Observational retrospective study	*n*: 4488 Mean age of those who died: 73 yrs	Tachycardia and ventricular fibrillation: 5	Death: 60 No clinical severity described WHO, 2009 Related to gravity	*n*: 13 Comorbidities: hypertension; diabetes mellitus; chronic kidney disease; cardiovascular
Lee et al., 2022 [66] Travel Medicine and Infectious Disease	Taiwan	Observational prospective descriptive study	*n*: 163 Median age: 72 yrs M: 25/F: 16	Troponin I: 82	Death: 21 Severe dengue with s-294 WHO, 2009 Related to gravity	*n*: 41 Comorbidities: hypertension; diabetes mellitus, kidney disease, ischemic heart disease
Wei et al., 2022 [43] PLos NTD	Taiwan	Population-based observation study Follow up: by 28 days	*n*: 65.906 Age: (Yrs) 0–39: 30 40–59: 237 ≥60: 977	Heart failure: 195	Dengue fever Secondary Dengue: not reported WHO: not reported	*n*: 844 Comorbidities: hypertension, diabetes mellitus, dyslipidemia Not related to gravity
Gupta et al., 2022 [44] Tropical Doctor	India	Observational prospective descriptive study Follow up: 1–22 days	*n*: 150 M: 100/F: 50 Mean age: 36 yrs	Myocarditis: 41 Sinus bradycardia: 11 Sinus tachycardia: 33 CK-MB: 43 Troponin I: 31	DF without warning: 41 DF with warning: 47 Severe dengue: 62 WHO, 2009	*n*: 41 Associated with dengue severity Comorbidities: not reported
**CHIKUNGUNYA**						
Obeyesekere et al., 1973 [32] Amer Heart Jour.	Sri Lanka	Observational study, prospective	*n*: 35 M: 17/F: 18 Age: 5–58 yrs	Cardiomegaly: 26 Cardiac arrhythmia: 25	Death: 3 (2-heart failure).	*n*: 26 Comorbidities: not reported
Lemant et al., 2008 [77] Crit Care Med	Reunion	Observational study, prospective	*n*: 33 Median age: 62 yrs M: 17/F: 16	Myocarditis: 1	Death: 16 Atypical forms: 19 (hepatitis, encephalopathy, shock, myocarditis)	*n*: 1 (myocarditis) Comorbidities: diabetes mellitus, cardiac failure Related to gravity
Rollé et al., 2016 [74] Emerging Infectious Diseases	Guadalupe	Prospective observational/ sample for convenience	*n*: 110 Median age: 71 yrs M: 62/F: 48	Cardio-circulatory failure: 22	Death: 14 (severe sepsis) Atypical forms: 34 Severe forms: 32 Non severe forms: 44	*n*: 38 Cardiac manifestations (10 non severe; 28 severe CHIKV) Comorbidities: DM; chronic heart disease; chronic renal disease Related to gravity
Bonifay et al., 2018 [78] PLoS ONE	French Guiana	Retrospective descriptive Follow up: median-5 days	*n*: 96 Median age-57 years	Cardio-respiratory failure (acute respiratory failure = 4, acute heart failure = 2	Death: 01 Common CHIKV-68 Atypical CHIKV-23 Severe CHIKV-5	*n*: 2 *Comorbidities*: fever, arthralgia headache, myalgia Related to gravity
Sharp et al., 2021 [75] Clinical Infectious Diseases	Puerto Rico	Retrospective descriptive sample for convenience	*n*: 30-fatal cases Median age: 61 yrs M: 19/F: 11	Cardiac arrhythmias: 11 Myocarditis: 1 Myocardial infarct: 4	Death: 30 Not reported	*n*: 16 Cardiac arrhythmias: 11 Myocarditis: 1 Myocardial infarct: 4 Comorbidities: 27 (DM; obesity, hypertension, coronary artery disease, asthma)
Lima et al., 2021 [76] Clinical Infectious Diseases	Brazil	Retrospective descriptive sample for convenience	*n*: 68 fatal cases Median age: 51 yrs F: 37/M: 31	Cardiac arrest: 23 Myocarditis: 15	Death: 68 Not reported	*n*: 38 Heart and/or respiratory failure were the most frequent causes of death. Comorbidities: hypertension; diabetes Related to gravity
**ZIKA**						
Cavalcanti at al., 2017 [80] PLoS One	Brazil	Observational retrospective study Follow up: not reported	*n*: 103 Mean age: 58 days	Clinical cardiologic- congenital heart disease - Ostium secundum: 5 - Small apical muscular ventricular septal defect.: 8	Clinical cardiologic-dyspnea	*n*: 14 Comorbities: not
Orofino et al., 2020 [79] Rev Inst Med Trop São Paulo	Brazil	Prospective observational study	*n*: 15 Median age: 16 months F: 6/M: 9	Not reported	Congenital Zika virus syndrome Severe microcephaly	The findings in the 24-h Holter monitoring suggest that infants with in utero exposure to ZIKV and severe ^###^ CZS are at higher risk of ^####^ SIDS

**Legend:** * DHF—Dengue Hemorrhagic Fever; ** DSS—Dengue Shock Syndrome; *** Dengue Fever ^#^ Ejection fraction; ^##^ Electrocardiograph; ^&^ Combined myocardial performance index. ^##^^#^ CZS—congenital Zika virus syndrome; ^###^^#^ SIDS—sudden infant death syndrome.

## Data Availability

The data supporting the reported data can be accessed via the link https://github.com/jdinicacio/arbovirusis.git.

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
