# Peer review of "Heart Disease and Arboviruses: A Systematic Review and Meta-Analysis"

_viruses, 2022, doi:10.3390/v14091988_

Round 1
Reviewer 1 Report
Manuscript ID: viruses-1853148
Comments to the Author:
General comments
In the present study, “Heart Disease and Arboviruses: A Systematic Review and Meta-Analysis”, Nicacio et al, performed a systematic review of multiple scientific databases to investigate the burden of cardiac complications related to dengue virus (DENV), Zika virus (ZIKV) and chikungunya virus (CHIKV). This is a very important and relevant topic considering the emergence of arboviral infections and the potential public health implications of arboviral-cardiac complications in endemic areas.
While several systematic reviews addressing the topic of arboviral infections and heart diseases have been made for CHIKV (i.e: PMID= 34206332, 28503297, and 33789296), DENV (i.e: PMID= 34292294), and ZIKV (i.e: PMID= 33220438 and, 28856072), this review provides a novel meta-analysis of DENV studies, which highlights that the higher incidence of dengue-related myocarditis was found in the population younger than 20 years old.
The strengths of this review are:
(i) Provides a comprehensive meta-analysis study of 17 dengue-related manuscripts.
(ii) Addresses cardiac complications associated with different arboviral diseases
While this review is complete from the DENV perspective, the “CHIKV-associated cardiac complications” and “ZIKV-cardiac complications” are not fully reviewed or discussed. Several relevant studies linking CHIKV and heart disease are not included (i.e: PMID= 18679124, 18694529, 27088710, and 35389992, to name a few).
In addition, important findings that are crucial to link CHIKV and heart disease, such as the identification of CHIKV antigen in cardiac tissue (PMID= 32615591), or heart-specific autopsy findings (PMID= 32766829) in CHIKV autopsies are not discussed in this study.
While this reviewer acknowledges the exclusion criteria stated for this study (non-original studies, reviews, editorials, abstracts, and animal model studies), major revisions of the CHIKV and ZIKV sections are required to further consider this manuscript for publication. In addition, the authors need to specify in the text what are the novel aspects that this review provides in comparison to the previously published reviews (i.e: PMID= 34206332, 28503297, and 33789296), DENV (i.e: PMID= 34292294), and ZIKV (i.e: PMID= 33220438 and, 28856072).
Point-by-point comments:
· Why case reports were not included in the study?
· Introduction: Please add/expand on typical and atypical manifestations due to infections with DENV, CHIKV, and ZIKV.
· Please add information regarding the evidence of direct heart infection for DENV, ZIKV, and CHIKV that support cardiac tropism.
· Line 81-82. Please change the reference to the most updated CHIKV epidemiological data https://www.ecdc.europa.eu/en/chikungunya-monthly.
· Line 87-89. Please modify the statement in lines (L87-89) since CHIKV is considered a highly symptomatic disease with only 15% of the cases presumably asymptomatic (PMID= 28159534).
· Line 100-102. Please modify the sentence in L100-102. WHO declared the ZIKV a public health emergency in 2016, not 2019.
· Line 118, the references selected are not representatives of the statement.
· Please explain in materials and methods how was done the scoring for the study selection. Which were the criteria to define the NOS for each study?
· It will be very informative to add to Table 1: (i) DENV serotype information and (ii) outcome (recovery, long-term cardiac complication, dead, or not available information) for the individuals with cardiac manifestations.
· Table 1. What does it mean ‘not related gravity’ in the “Cardiac involvement” column?
· L253-L260. Analysis of ZIKV and CHIKV cardiac complications needs to be expanded.
· L317. Replace “Arboviruses are acute infections” with “arboviruses are associated with/can cause acute infections”
· Line 320-322. While the data shows that cardiac complications were found in over a quarter of infected participants, the authors must acknowledge that the studies focused on hospitalized individuals.
· L355-356. Replace “arbovirus-related” with “DENV-related”. So far, the authors have shown this finding only for DENV.
Reviewer 2 Report
Retrospective review of reported cases is always creditable. This review is a great effort in studying the relationship between heart disease and arboviruses. Unfortunately, restricted by the quality of primary studies, the cardiac involvement in arboviruses infection is not solid, however it opened the door for future studies.
One particular question: among all the studies, do you have the information of vaccine protection? If some infections were recorded after vaccination, did the vaccination protect the the patient from cardiac involvement?
Round 2
Reviewer 1 Report
The authors have addressed all my concerns.